# Transcriptome Analysis of Induced Pluripotent Stem Cells and Neuronal Progenitor Cells, Derived from Discordant Monozygotic Twins with Parkinson’s Disease

**DOI:** 10.3390/cells10123478

**Published:** 2021-12-09

**Authors:** Ivan N. Vlasov, Anelya Kh. Alieva, Ekaterina V. Novosadova, Elena L. Arsenyeva, Anna V. Rosinskaya, Suzanna A. Partevian, Igor A. Grivennikov, Maria I. Shadrina

**Affiliations:** 1Institute of Molecular Genetics of National Research Centre, Kurchatov Institute, 2 Kurchatova Sq., 123182 Moscow, Russia; anelja.a@gmail.com (A.K.A.); novek-img@mail.ru (E.V.N.); arslena@mail.ru (E.L.A.); spartev@img.ras.ru (S.A.P.); igorag@img.ras.ru (I.A.G.); shadrina@img.ras.ru (M.I.S.); 2State Public Health Institution Primorsk Regional Clinical Hospital No. 1, 57 Aleutskaya St., 690091 Vladivostok, Russia; rosinskaya@bk.ru

**Keywords:** transcriptome, Parkinson’s disease, monozygotic twins, differentiation, neuronal progenitor cells, induced pluripotent cells

## Abstract

Parkinson’s Disease (PD) is a widespread severe neurodegenerative disease that is characterized by pronounced deficiency of the dopaminergic system and disruption of the function of other neuromodulator systems. Although heritable genetic factors contribute significantly to PD pathogenesis, only a small percentage of sporadic cases of PD can be explained using known genetic risk factors. Due to that, it could be inferred that changes in gene expression could be important for explaining a significant percentage of PD cases. One of the ways to investigate such changes, while minimizing the effect of genetic factors on experiment, are the study of PD discordant monozygotic twins. In the course of the analysis of transcriptome data obtained from IPSC and NPCs, 20 and 1906 differentially expressed genes were identified respectively. We have observed an overexpression of *TNF* in NPC cultures, derived from twin with PD. Through investigation of gene interactions and gene involvement in biological processes, we have arrived to a hypothesis that *TNF* could play a crucial role in PD-related changes occurring in NPC derived from twins with PD, and identified *INHBA*, *WNT7A* and *DKK1* as possible downstream effectors of *TNF.*

## 1. Introduction

Parkinson’s Disease (PD) is a severe neurodegenerative disease that affects one to two individuals per thousand in the total population, and approximately 1% of the population over 60 years of age [1]. Pronounced deficiency of the dopaminergic (DA) system and disruption of the function of other neuromodulator systems are crucial traits of PD [2,3]. The classic motor symptoms of PD include bradykinesia, tremor, muscle rigidity, and postural instability. In addition to motor symptoms, patients suffering from PD can experience hyposmia, sleep disturbance, depression, and dysautonomic symptoms such as constipation [4]. These symptoms develop against the background of an inevitable progression to the death of DA neurons in the substantia nigra. Non-motor symptoms usually develop at earlier stages of PD progression, whereas motor symptoms only appear after the death of over 50% of DA neurons and a 70–80% reduction of dopamine levels in the striatum [5,6]. 

It is widely known that heritable genetic factors, such as singular rare highly penetrant variants or risk factors identified in association studies, contribute significantly to PD pathogenesis [7,8,9,10]. Although most cases of PD are considered to be sporadic, only a small percentage of sporadic cases of PD can be explained using known genetic risk factors [8]. 

Due to that, it could be inferred that changes in gene expression could be important for explaining a significant percentage of PD cases. One of the ways to investigate such changes, while minimizing the effect of genetic factors on experiment, are the study of PD discordant monozygotic twins.

Few studies have investigated molecular markers in discordant monozygotic twins with PD. In a study published in 2014, Woodard et al. measured neuronal differentiation efficiency, GBA enzyme activity, and alpha-synuclein levels in cells derived from discordant twins with PD carrying an N370S mutation in the *GBA* gene. That study demonstrated that the N370S mutation inhibits the transport of alpha-synuclein out of the cell [11]. In a study reported in 2017, Kaut et al. analyzed genomic methylation in a cohort of 62 discordant siblings with PD, including 24 monozygotic twins. Overall, 62 differential methylation sites in proximity of 15 genes were identified in the twins. Concomitantly, during validation using an independent population of patients with PD, only three genes were identified as carrying multiple confirmed differential methylation sites: *TRIM34*, *PDE4D*, and *MIR886* [12]. In a 2020 study by Mazetti et al., the authors investigated the accumulation of oligomeric alpha-synuclein in skin biopsies of patients with sporadic PD and in 19 pairs of discordant monozygotic twins with PD. It was confirmed that oligomeric alpha-synuclein accumulates in the synaptic terminals of autonomous nerve fibers in twins suffering from PD [13]. In 2021, Dulovic-Mahlow et al. conducted full genome sequencing, deep mitochondrial DNA sequencing, and mitochondria integrity assessment on fibroblasts derived from five pairs of discordant twins with PD. In that study, changes in mitochondrial morphology, upregulation of the SOD2 protein, and a reduction in cellular ATP levels were observed in the fibroblasts derived from twins with PD [14]. As evidenced in the studies mentioned above, no transcriptomic profiling of gene expression in discordant twins with PD has been conducted. Therefore, such an investigation appears to be a novel, important direction of research in this field.

In our previous study, we analyzed the transcriptomes of fibroblasts derived from discordant twins with PD. We identified common differentially expressed genes (DEGs) between discordant twins with PD, including genes related to biological processes such as action potential, glutamatergic synaptic transmission, and adipocyte differentiation [15].

In the present study, we performed a transcriptomic analysis of induced pluripotent stem cells (IPSCs) and neuronal progenitor cells (NPCs) derived from fibroblasts collected from discordant twins with PD.

## 2. Materials and Methods

### 2.1. Discordant Monozygotic Twins with PD

Two pairs of phenotypically and genetically discordant monozygotic twins with PD were enrolled in the study. These individuals (of Russian ethnic origin and residing in European Russia) were diagnosed at the State Public Health Institution Primorsk Regional Clinical Hospital No. 1 and did not have any family history of PD. The participants were assessed according to the International Parkinson and Movement Disorder Society-sponsored Unified Parkinson’s Disease Rating Scale [16] and Hoehn–Yahr scores [17]. The diagnosis of PD was based on the UK Parkinson’s Disease Society Brain Bank criteria [18]. Patients were at stages 2–4 of the Hoehn–Yahr scale and had a mixed form of PD. The disease duration was at least 7 years. The healthy siblings did not have any signs of PD at the time of collection of the biological material. The twins lived in the same area, and their work was not associated with dangerous factors, such as pesticides or heavy metals. There was no information about the presence of head injuries in their history [15]. Sex, age, and stage of the disease are presented in Appendix A. All patients were pre-typed using the P051-50 (lot C2-0911) and P052-50 (lot C1-0809) probe sets and EK1-FAM reagent kit for the SALSA MLPA MLPR (MRC-Holland, Amsterdam, The Netherlands) according to the manufacturer’s recommendations; none of them had any frequent mutations associated with pathogenesis of the PD.

### 2.2. Skin Biopsy and Obtaining the Primary Culture of Skin Fibroblasts

The fibroblasts used in this study were obtained previously as described in reference [15].

### 2.3. IPSC Reprogramming and Cultivation

IPSC reprogramming was achieved using a non-integrational Sendai virus (CytoTune-iPS 2.0 Sendai Reprogramming Kit, Invitrogen, Waltham, MA, USA), in accordance with the manufacturer’s recommendations. Mouse embryonic fibroblasts (MEFs) were used as a substrate. Live-cell TRA-1-60 antibody staining was carried out 21–29 days after IPSC colony emergence using a mouse anti-human mAb and an AlexaFluor 488 Conjugate Kit for Live Cell Imaging. TRA-1-60 is a surface protein that is considered to be a pluripotency marker. Staining was used to identify individual clones for further propagation. Stained colonies were mechanically transported to a 24-well plate containing a prepared substrate of inactivated MEFs. To confirm the success of IPSC cell reprogramming, cell lines were examined for the presence of pluripotency markers and the ability to form three germ layers. The obtained IPSC lines were cultivated on Matrigel (BD Biosciences, San Jose, CA, USA) substrate in mTeSR medium (STEM CELL Technologies, Vancouver, Canada) at high humidity, 37 °C, and 5% CO_2_. Cells were passaged 1:2 or 1:3 every 5–6 days using 1 mg/mL of dispase (Invitrogen, Waltham, MA, USA). During reseeding, a ROCK inhibitor (StemoleculeY27632, Stemgent, Beltsville, MD, USA) was added to the medium at a final concentration of 5 μM.

### 2.4. Differentiation and Cultivation of Neuronal Progenitor Cells

After IPSCs had reached the monolayer stage, the mTeSR medium (STEM CELL Technologies, Vancouver, Canada) was replaced with neuronal differentiation medium ((DMEM/F12 (Gibco, Waltham, MA, USA), 2% serum replacement (Gibco, Waltham, MA, USA), 1% N2 additive (Gibco), 2 mM glutamine (ICN Biomedicals, Costa Mesa, CA, USA), 1% amino acid mix (PanEco, Moscow, Russia), 50 U/mL of penicillin–streptomycin (PanEco, Russia), 80 ng/mL of Noggin (PeproTech, East Windsor, NJ, USA), and 10 μM SB431542 (Stemgent, Beltsville, MD, USA)). Cells were cultivated in this medium for 12–14 days. Subsequently, cells were reseeded on Matrigel-treated dishes (BD Biosciences, San Jose, CA, USA) in neuronal differentiation medium with addition of 5 μM ROCK inhibitor (StemoleculeY27632, Stemgent, Beltsville, MD, USA). NPCs collected after three to five passages were used for RNA isolation.

### 2.5. Embryoid Bodies Formation

IPSC colonies were detached using dispase (Gibco, Waltham, MA, USA). In order to do that, the culture medium was removed, cells were rinsed using DMEM medium with 1 mL of added dipase (1 mg/mL) per 35 mm dish, incubated at 37 °C for 7–10 min. Then dispase solution was removed and dish was rinsed 5 times with 1 mL of DMEM medium. After that 1 mL of the mTeSr culutre medium was added.

Colonies were scraped off using 200 μL plastic tip. After scraping off they were carefully dissociated in 400–600 cell fragments and transferred into 24-well Ultra low adhesion plates (Costar, Moscow, Russia, Ultra-Low Attachment Surface). Next day, when IPSC have formed embryoid bodies (EB) half of the volume of medium were replaced with EB cultivation medium (DMEM/F12 (Gibco, Waltham, MA, USA), 20% FBS (Hyclone, Logan, UT, USA), 2 mM L-glutamin (ICN Biomedicals, Costa Mesa, CA, USA), 0.1 mM β-mercaptoethanol (SIGMA, Saint Louis, MO, USA), 1% mix of non-essential amino acids (PanEco, Moscow, Russia), penicillin-streptomycin (PanEco, Moscow, Russia) (50 μL/mL). This partial replacement was conducted daily, until the cultural medium was completely replaced with EB cultivation medium. After that medium was replaced bi-daily.

### 2.6. Spontaneous IPSC Differentiation into Derivates of Three Germ Layers

To obtain meso-, ecto- and ento-dermal derivates of germ layers 3–4 days old EB were transferred to gelatine coated Petri dishes with the EB cultivation medium. EB were cultivated for 21 days. Medium was replaced every 48 h. On 22 day of cultivation cells were fixed using 4% paraformaldehyde (SIGMA, Saint Louis, MO, USA),

### 2.7. Immunocyto Chemical Staining of Cell Cultures

Cells were rinsed with PBS solution (ICN Biomedicals, Costa Mesa, CA, USA) and fixed in 4% paraformaldehyde (SIGMA, Saint Louis, MO, USA) for 20 min at room temperature. After fixation they were rinsed three more times in PBS solution and incubated in PBS—0.1% Triton x-100—5% serum for 15 to permeabilize them and reduce the non-specific anti-body sorbtion.

Following that, cells were incubated in PBS—0.1% Triton x-100—5% solution with specific anti-bodies for a night at +4 °C. Then cells were rinsed in PBS—0.1% Tween 20 solution three times. After that they were incubated in PBS—0.1% Triton x-100—5% with secondary anti-bodies, stained with fluorescent marker for 1.5 h at room temperature.

Following incubation with secondary antibodies they were again rinsed three times in PBS—0.1% Tween 20 solution, they were incubated in solution of 0.1 μg/mL solution of DAPI (SIGMA, Saint Louis, MO, USA) for 1 min, and then once again rinsed three times in PBS—0.1% Tween 20 solution.

Results were visualized using AxioImage (ZEISS, Oberkochen, Germany) microscope.

### 2.8. Primary Anti-Bodies

To identify expression of pluripotency markers following antibodies were used (Abcam, Cambridge, MA, USA):-rabbit polyclonal Anti- OCT4-mice monoclonal Anti-SSEA-4

To identify the potency os IPSC to differentiate into derivatives of three germ layers following antibodies were used (Abcam, Cambridge, MA, USA):-mice monoclonal anti-Sox1—ectodermal marker-rabbit polyclonal anti-desmine—mesodermal marker-mice monoclonal anti-AFP—entodermal marker

To confirm the differentiation into NPC following antibodies were used (Abcam, Cambridge, MA, USA):-mice monoclonal Anti-Sox1

### 2.9. Secondary Antibodies

Following antibodies were used as secondary:-goat anti-mice Alexa Fluor 488, (A11001) 1/2000-goat anti-rabbit Alexa Fluor 546, (A11010), 1/2000-goat anti-chicken Alexa Fluor 546, (A11040), 1/3000 (Invitrogen, Waltham, MA, USA)

### 2.10. Karyotype Analysis

Karyotype of iPSCs was defined using G-banding at a resolution of 400 bands with 20 metaphase plates being analyzed. Colcemid was used in the final concentration of 0.2 μg/mL. Metaphases were scored using a Metafer semi-automated system and IKAROS software (MetaSystems GmbH, Altlussheim, Germany).

### 2.11. RNA Isolation and Sequencing

Three replications for each sample and each cell culture (2 patients × 2 conditions × 2 cultures × 3 replicates = 24 total) were used for isolation and sequencing. RNA was isolated using Trisol (Invitrogen, Waltham, MA, USA) in accordance to manufacturer’s recommendations.

Quality and quantity of isolated RNA was evaluated using «BioAnalyser» tool and RNA 6000 Nano Kit (Agilent, Santa Clara, CA, USA). Poly(A) fraction was extracted from total RNA for sequencing. Libraries for sequencing were prepared from poly(A) fraction using NEBNext^®^ mRNA Library Perp Reagent Set (NEB, Rowley, MA, USA). Sequencing was performed using HiSeq1500 (Illumina, San Diego, CA, USA), obtaining no less than 10 mln. 50 bp reads per library.

### 2.12. RNA-Seq Data Analysis

FASTQ file, generated during sequencing were trimmed for ambiguous and low quality bases using AdapterRemovalV2 [19]. Transcriptome for mapping was generated using human genome GRCH38 and gene annotation GRCH38.92 using RSEM [20] command rsem-prepare-reference with –star option enabled to also generate STAR indices [21].

Mapping was performed using STAR and RSEM, command rsem-calculate-expression with —star option enabled. Obtained pseudocounts were normalized using TMM algorithm, implemented in command «calcNormFactors» from R package «edgeR», [22] and CPM algorithm, implemented in “voom” command from R package “limma” [23].

To identify the differential expression, normalized reads were processed using commands “voom” (estimating mean/dispersion ratio, calculating observation weights), “lmFit” (fitting a linear model for contrasts) and “eBayes” (calculating parameters of the linear model) from R package “limma” [23].

Genes were considered to be differentially expressed based on criteria of FC > 1.5 and *p*-value of moderated *t* test from limma with FDR correction < 0.05.

### 2.13. GO BP Term Enrichment

Gene Ontology Biological Processes (GO BP) [24] term enrichment was conducted using apps ClueGO v. 2.5.3 [25] and Cluepedia v. 1.5.3 [26] for Cytoscape v. 3.6.1.

Significantly enriched terms were selected based on one sided hypergeometric tests with Bonferroni correction (corrected *p*–value < 0.001). Term groups were formed based on common genes per term (>40%). Only GO BP terms of higher than 4th level with at least 3 DEGs associated with them were considered for enrichment. Also terms, for which at least 10% of total associated genes were not differentially expressed were not considered for enrichment.

Visualization of relations between terms and associated genes was performed by ClueGO v. 2.5.3 and Cluepedia v. 1.5.3.

### 2.14. Network of Interaction of Genes and Metabolic Processes

Gene and metabolic process interaction networks were created using Pathway Studio^®^ v. 12.1.0.9 (Elsevier, Amsterdam, The Netherlands). While creating networks only interactions, supported by at least 2 sources, were considered.

## 3. Results

### 3.1. Cell Culture Characterization

Cell culture characterization can be found in Appendix A.

Whole-transcriptome analyses of IPSCs and NPCs obtained from two pairs of discordant twins with PD were performed. A comparison of the levels of gene expression between the twins who were healthy and those who had PD was carried out, and DEGs were identified.

### 3.2. IPSC Transcriptome Analysis

In the course of the IPSC transcriptome analysis, 20 DEGs (DEG IPSC) were identified. A full list of DEG IPSC is provided in Appendix A. Top 3 genes up- and down-regulated by fold change are provided in Table 1. A Gene Ontology Biological Processes (GO BP) enrichment was conducted for DEG IPSC; however, no significantly enriched terms were identified.

Gene interaction networks were built based on PW Studio information about the interaction of DEGs and PD, their expression in neurons and neural stem cells, and their interaction with biological processes, as identified using the key words “neurodegeneration”, “transport”, “vesicular”, “mitochondria”, “lysosome”, “oxidative stress”, “apoptosis”, and “autophagy”. Four DEGs (*NR2F2*, *P2RY12*, *DDX43*, and *BLNK*) were connected to aforementioned biological processes: two were connected to neurons (*NR2F2* and *P2RY12*) and one to PD (Figure 1).

### 3.3. NPC Transcriptome Analysis

In the course of the NPC transcriptome analysis, 1906 DEGs (DEG NPC) were identified. A full list of DEG NPC is provided in Appendix A. Top 5 genes up- and down-regulated by fold change are provided in Table 2.

The obtained DEG NPC were used for GO BP enrichment. Significantly enriched terms were grouped based on the percentage of common genes (>40% common genes). Three groups consisting of 12 significantly enriched terms were identified (Table 3, Figure 2).

Groups I and II were identified as being most promising for further investigation because they are directly linked to neuronal differentiation. An additional analysis of these groups was performed (Figure 3). A total of 90 DEGs were associated with at least two terms, including 27 associated with terms from groups I and II simultaneously, as well as 23 associated with at least two terms from group I and 40 with at least two terms from group II.

Twenty-seven DEG NPC associated with terms from groups I and II were selected for further investigation. An interaction network was built in Pathway Studio for these genes using key words such as “Parkinson”, “neurodegeneration”, “transport”, “vesicular”, “mitochondria”, “lysosome”, “oxidative stress”, “apoptosis”, and “autophagy” (Figure 4). For visual clarity regarding processes and genes, those connected to only a single or no other genes were removed from the analysis. Excessive and redundant terms and diseases not connected to PD were also removed.

Twenty of 27 genes interacted with PD or the aforementioned biological processes. Those genes were selected for further investigation as the most promising candidates from the standpoint of their connection to PD.

Six genes were identified as being differentially expressed in the same direction in both NPCs and IPSCs: *DDX43*, *P2RY12*, *ZNF729*, *LINC02864*, *AC018521.1*, and *AC090498.1.*

An interaction network of the previously identified most promising DEG IPSC (DEG in Figure 1) and DEG NPC (DEG in Figure 4) was constructed (Figure 5). After identifying all interactions between genes, PD, and neurodegeneration, all genes that did not exhibit any interaction with at least one other gene or term were removed from the analysis. As a result, two DEG IPSC (*P2RY12* and *NR2F2*) and nine DEG NPC (*P2RY12*, *TNF*, *PITX3*, *GRID2*, *INHBA*, *DLL4*, *WNT7A*, *DKK1*, and *FGF19*) were identified. *NR2F2* has been shown to interact with *TNF* and *DLL4*, whereas *P2RY12* interacts only with *TNF.*

## 4. Discussion

Despite the fact that PD mostly affects the central nervous system (CNS), PD pathogenesis can also affect peripheral tissues and non-neuronal cells [27]. This is also consistent with the results obtained in our previous work, which demonstrated significant differences between the expression profiles obtained from the fibroblasts of discordant monozygotic twins with PD [15]. One of the explanations for this observation would be that those differences are the result of subtle changes that occur at the earliest states of disease ontogenesis. To test this hypothesis, a transcriptomic profiling of IPSCs and NPCs differentiated from IPSCs obtained from discordant monozygotic twins with PD was performed.

In the course of the transcriptomic profiling of IPSCs, 20 DEGs (DEG IPSC) were identified between the twins who were healthy and those who had PD. No significantly enriched GO BP were identified for DEG IPSC. Thus, no involvement of metabolic processes at the genomic level could be identified. In the next step, investigations of the interactions between DEG IPSC, PD, neurons and neuronal stem cells, and various PD-relevant biological processes were performed using Pathway Studio (Figure 1). We detected interactions of four of 20 DEG IPSC with biological processes that are important for PD, such as apoptosis, oxidative stress, and mitochondrial damage. Moreover, a direct interaction between *NR2F2* and PD was identified. An additional literature analysis identified interactions between those four genes and PD, thereby uncovering a possible connection between *P2RY12* and PD [28,29]. Interactions between *NR2F2* and *P2RY12* and neurons have been also identified. Because of the potential link between PD, neurons, *NR2F2*, and *P2RY12*, these two genes were identified as being most promising for further investigation.

*P2RY12* encodes the purinergic receptor P2RY12. *P2RY12* is expressed in a variety of tissues [30,31]. This receptor is also expressed during CNS development, starting from the embryonic stage [32]; moreover, in the adult CNS, it is expressed in microglia [32,33]. The P2RY12 protein can be considered a marker of activated microglia, as its expression is reduced during neuroinflammation [32].

Several studies have demonstrated a putative relationship between *P2RY12* and PD. Using a meta–analysis of genome-wide association study data, Andersen et al. identified a significant association between *P2RY12* and PD [29]. Using machine learning models, Shen et al. found that changes in the expression of *P2RY12* were a predictor of PD [28]. 

In general, such associative data do not allow the understanding of how a specific gene is involved in the pathogenesis of a particular disease. To elucidate the potential role of genes in PD pathogenesis, the interactions between genes and biological processes with known relation to PD, such as apoptosis [34,35,36], autophagy [34,35], vesicular transport [37], mitochondria damage [38] and oxidative stress [39], have been investigated.

Currently, the data linking *P2RY12* to apoptosis are ambiguous. Sun et al. demonstrated that the activation of P2RY12 is a pro-apoptotic factor [40]; conversely, Mamedova et al. showed that the activation of P2RY12 could be an anti-apoptotic factor [41]. Nevertheless, these two studies can be taken as evidence that *P2RY12* is a regulator of TNF-alpha-mediated apoptosis.

Furthermore, it has been demonstrated that the suppression of *P2RY12* expression leads to autophagy-mediated cell death [42] and to the suppression of oxidative stress during inflammation [43].

Thus, data currently exist that point to a link between *P2RY12* and PD and the regulation of PD-relevant processes such as apoptosis, autophagy, and oxidative stress. Therefore, its differential expression in IPSCs obtained from discordant monozygotic twins with PD is probably specific to PD and may reflect processes occurring in the CNS, particularly in microglia.

The *NR2F2* gene encodes an orphan nuclear receptor that is an important differentiation regulator and plays a role in the homeostasis of various tissues. Despite the fact that its expression peaks during the embryonic stage of development, changes in its expression could be linked to various diseases [44].

Other sets of data suggest that overexpression of *NR2F2* could be associated with PD. In several works involving PD models and tissues derived from patients with PD, its overexpression was identified in the substantia nigra [45] and DA neurons [46,47,48,49,50]. In addition, the association between *NR2F2* overexpression and oxidative stress has been demonstrated [49]. The important role that oxidative stress plays in the pathogenesis of PD suggests that *NR2F2* is linked to PD through oxidative stress.

Despite the fact that the literature implies a link between the increased expression of *NR2F2* and PD, in our data, a significant decrease in the expression of this gene was identified in IPSCs derived from twins with PD. A possible explanation for this fact could be that the downregulation of *NR2F2* is a compensatory mechanism that occurs in response to oxidative stress. This hypothesis is supported by the fact that a reduction in *NR2F2* expression in vivo slows down the progression of motor symptoms and neurodegeneration and supports the level of dopamine in the striatum [49]. Thus, the observed decrease in the expression of *NR2F2* in IPSCs derived from twins with PD could be related to a compensatory reaction to oxidative stress in this disease.

In the course of the analysis of transcriptome data obtained from NPCs, 1906 DEGs (DEG NPC) were identified. It is noteworthy that this represents a significant increase compared with that observed for IPSCs: from 20 DEG IPSC to 1906 DEG NPC (Appendix A). It is also noteworthy that such significant differences between IPSCs and NPCs cannot be related to genomic differences, as the cell lines were derived from monozygotic twins. A possible explanation for this observation would be the involvement of additional genes regulated by DEG IPSC during differentiation. First, one or more DEG IPSC themselves could be “master regulators” of genes that are enabled at the NPC level and directly regulate the expression of multiple NPC genes. Alternatively, DEG IPSC could regulate a “master regulator” of NPC genes. It is noteworthy that *NR2F2* was not differentially expressed at the NPC level, which precludes it from acting as a putative “master regulator”. To explore which genes could act as this putative master regulator of NPC genes and cause such massive increases in the number of DEG NPC compared with DEG IPSC, we analyzed the functions, interactions, and relationships to DEG IPSC of DEG NPC further.

The GO BP enrichment analysis identified three groups of biological processes that were significantly enriched among DEG NPC (Figure 2). Group I included processes related to neurogenesis and the differentiation of neurons; group II encompassed processes related to the morphogenesis of anatomical structures, particularly the neural tube; and group III included processes related to biosynthetic and metabolic regulation. The most promising entities for further investigation were the genes that were involved in both group I and group II (Figure 3) because they are directly related to differentiation and their presence implies that neuron-predecessor differentiation is proceeding in different directions between the twins who are healthy and those who have PD. Overall, 27 such genes were identified. The links between these genes and PD and biological processes that are important for PD were investigated (Figure 4), and as a result, 20 genes exhibiting such links were identified.

As mentioned previously, the increased divergence between healthy and PD transcriptomes from IPSCs to NPCs could be explained by the existence of a “master regulator” and/or regulator of the “master regulator” of NPC genes among DEG IPSC. To investigate this hypothesis, links between DEG IPSC related to PD and biological processes important for PD (Figure 1) and the DEG NPC involved in groups I and II of GO BP and related to PD and biological processes significant for PD (Figure 4) were investigated. As a result, an interaction network of those genes was constructed (Figure 5).

*NR2F2* and *P2RY12* are the only genes among DEG IPSC for which a link to PD has been previously identified. These genes are only connected to DEG NPC through *TNF* and *DLL4* (Figure 5), which is incongruent with the hypothesis of them being “master regulators” of DEG NPC related to PD. In turn, *TNF* was linked to five of eight genes in the interaction network, which renders it a better candidate for the role of putative “master regulator” of DEG NPC related to PD.

Currently, data exist that suggest a connection between *NR2F2*, *P2RY12*, and *TNF*. As mentioned above, *P2RY12* is involved in TNF-mediated apoptosis [40,41]. Moreover, it has been demonstrated that *P2RY12* regulates the expression of several pro-inflammatory cytokines, including TNF, in microglial cell culture [51]. Yi et al. reported a co-dependent and co-directed change in the expression of *P2RY12* and *TNF* [52].

Data also exist that link *NR2F2* and *TNF*. In 2014, Litchfield et al. demonstrated that the overexpression of *NR2F2* leads to the downregulation of TNF-alpha-induced NFκB [53]. Moreover, the introduction of TNF-alpha led to a decrease in *NR2F2* expression in endometrial stroma [54]. These data imply that *NR2F2* suppresses TNF-mediated signaling, and that *TNF* in turn suppresses *NR2F2* expression.

The *TNF* gene encodes the pro-inflammatory cytokine TNF, also known as TNF-alpha, which plays a major role in multiple diseases and conditions, including neurodegenerative diseases such as PD and Alzheimer’s disease [55].

Currently, a large body of data exists in support of a connection between *TNF* and PD [55]. Some of the most significant points among those data are the studies that demonstrated that the suppression of TNF expression leads to a reduced risk of PD [56,57], a reduction of DA neuron loss in the substantia nigra, and the amelioration of neuroinflammation [58]. Furthermore, several works revealed an association between *TNF* polymorphisms and the risk of developing PD [59,60,61,62,63]. Increased levels of TNF have been detected in the spinal fluid [64], blood serum [65] and tears [66] of patients with PD.

Therefore, the correlation between *TNF* overexpression and PD development appears to be undoubtable. Our data also support this relationship, as *TNF* expression was increased in NPCs derived from twins with PD. The importance of *TNF* overexpression and its potential role as a “master regulator” of DEG NPC related to PD is supported by its involvement in both groups I and II of GO BP terms (Figure 3), and is linked to the higher number of DEGs with connections to PD and biological processes important for PD (five of 20) observed in these groups (Figure 5).

DEG NPC were linked to TNF, PD, and biological processes important for PD, and were involved in groups I and II of GO BP processes. The relation between those genes and *TNF* will be reviewed below. As mentioned previously, increased expression of *TNF* was observed in NPCs obtained from twins with PD. Considering our hypothesis about the role of *TNF* as a “master regulator” of the PD-related DEG NPC, we assumed that the expression of those genes is defined by their relation to *TNF*. In cases in which *TNF* inhibits the expression of these genes, their expression should be reduced in twins with PD in our data. In turn, the opposite should be observed in cases in which *TNF* induces the expression of those genes (Table 4).

Table 2 shows that the direction of the expression change observed in our data is in full accordance with that expected from literature data only for two genes, *INHBA* and *WNT7A*. It is worth mentioning that an involvement in PD pathogenesis has been demonstrated previously for these genes. We have also created barplots with by normalized counts by sample in order to investigate if differential expression is driven by a single differentiation or a distinct expression in a single twin (Appendix A).

The *INHBA* gene, which encodes the activin-A protein (a member of the TGF-beta factor family), plays a role in biological processes such as inflammation, fibrosis, and immunoregulation [72]. As mentioned previously, there exist data linking this gene to PD. Activin-a introduction leads to a significant increase in the survivability of DA neurons and neurons of the substantia nigra in a mouse model of PD, as assessed based on 6-OHDA injections [80]. Activin-A also has a neuroprotective effect in other forms of neurodegeneration. For instance, the introduction of activin-A yielded neuroprotective effects in a rat model of Huntington’s disease [95]. Moreover, intake of activin-A caused amelioration of neurogenesis after neurodegeneration caused by lipopolysaccharide injection in a mouse model [96].

*WNT7A* encodes the secreted protein Wnt7a, which is an activator of the canonical and non-canonical Wnt signaling pathways. This protein in particular and the Wnt pathway as a whole are involved in multiple biological processes, including differentiation, proliferation, wound healing, and inflammation suppression [81,82].

Moreover, data exist that suggest that this signaling pathway is involved in the pathogenesis of PD [85].

Table 2 also shows that the data pertaining to the interaction between *DKK1* and *TNF* are ambiguous. The *DKK1* gene encodes the DKK1 protein, which is an antagonist of *WNT7A* and the canonical Wnt pathway. WNT7A inhibits its expressing [86,97], and DKK1 inhibits WNT7A-mediated signaling [98]. Several studies have demonstrated that *TNF* intake upregulates DKK1 in several human cell cultures [89,90,91,92,93] and inhibits DKK1 expression in others [87,88]. The upregulation of *WNT7A* observed in our data could also affect the expression of DKK1. The upregulation of *TNF* and *Wnt7a* and the downregulation of *DKK1* observed in NPCs derived from twins with PD support this contention. The differential expression of *DKK1* and *WNT7A* points to a possible role for the Wnt pathway in the pathogenesis of PD.

## 5. Conclusions

Based on the data obtained here, we suggest a role for TNF as a “master regulator” of PD-involved DEG NPC. The possible mediation of the differential expression of TNF in NPCs by the differential expression of NR2F2 and/or P2RY12 could explain the divergence between the transcriptomic profiles of healthy and PD twins in the process of NPC differentiation. Overall, the data obtained here suggest the paramount importance of changes in the expression of TNF in PD pathogenesis.

Involvement of DEG in the process of differentiation from IPSC to NPC through *TNF* as an NPC DEG master regulator demonstrates a possible mechanism of exacerbation of minor differences in the process of tissue differentiation. Further studies on monozygotic twins, including more stages of tissue differentiation are required to further elucidate that mechanism.

Limitations: differences in gene expression could be caused by mutations and genetic variations, or by epigenetic changes. Due to the fact that in this paper we are examining monozygous twins, the most likely explanation of differential expression is epigenetic changes. However, cellular reprogramming necessarily affect the epigenetic markers within cells. Therefore, there inevitably exists a risk that differences in obtained transcriptomic profiles are not entirely representative of differences in transcriptomic profiles of original tissues in twins, discordant by PD.

## Figures and Tables

**Figure 1 cells-10-03478-f001:**
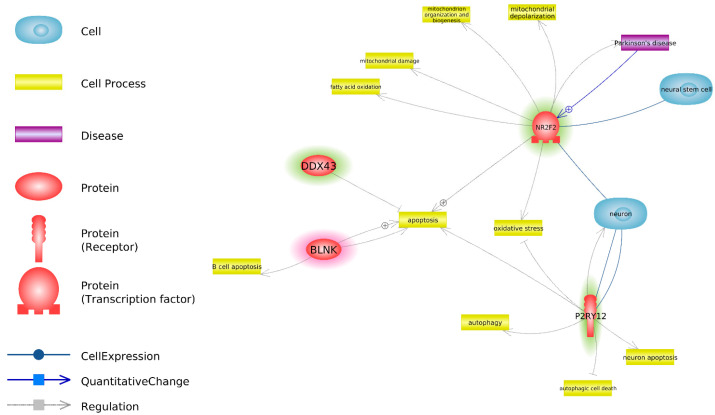
DEG IPSC interaction with neurons, neural stem cells, PD and biological processes, identified using key words “neurodegeneration”, “transport”, “vesicular”, “mitochondria”, “lysosome”, “oxidative stress”, “apoptosis” and “autophagy” according to Pathway Studio data. Genes, overexpressed in IPSC derived from twins with PD, as compared to IPSC derived from healthy twins are highlighted red. Genes, underexpressed in IPSC derived from twins with PD, as compared to IPSC derived from healthy twins are highlighted green.

**Figure 2 cells-10-03478-f002:**
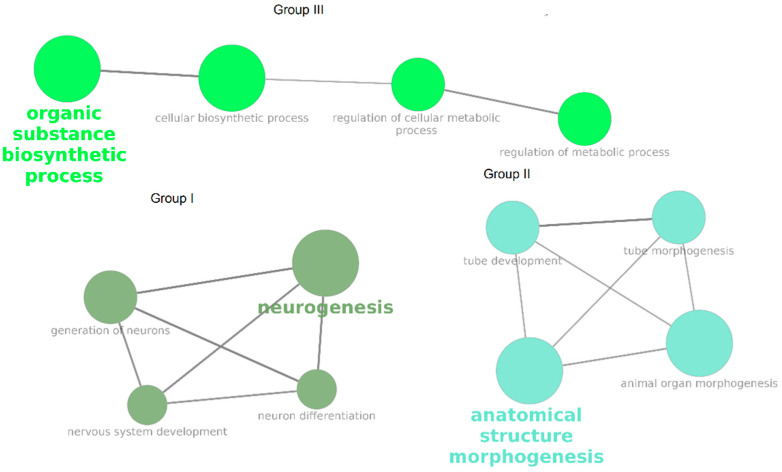
Significantly enriched GO BP terms for DEG NPC. Color denotes the grouping of terms based on percentage of common genes.

**Figure 3 cells-10-03478-f003:**
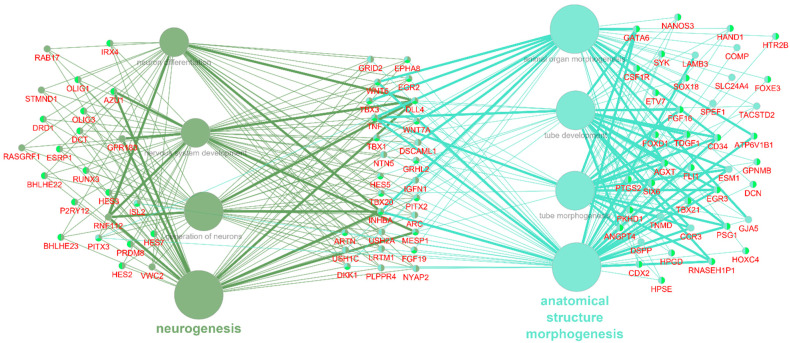
GO BP terms from groups I and II, significantly enriched with DEG NPC with DEG, associated with at least 2 terms. Genes in the middle are associated with terms from groups I and II. Genes on the left are associated with terms from group I, genes on the right are associated with terms from group II.

**Figure 4 cells-10-03478-f004:**
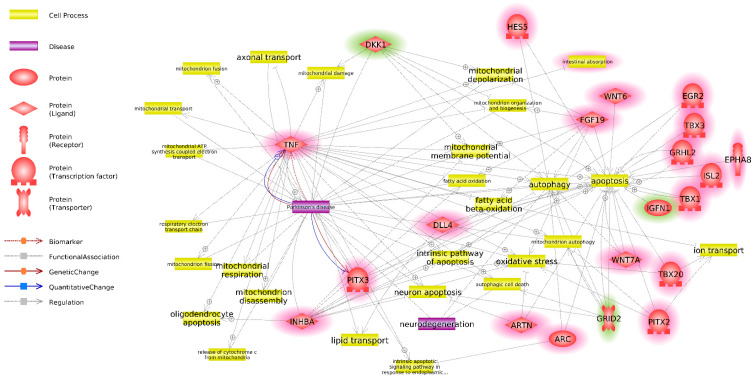
Interaction network of DEG NPC, associated with significantly enriched GO BP terms from groups I and II and diseases and metabolic processes, identified using key words “Parkinson”, “neurodegeneration”, “transport”, “vesicular”, “mitochondria”, “lysosome”, “oxidative stress”, “apoptosis” and “autophagy” based on Pathway Studio data. Genes, overexpressed in NPC derived from twins with PD, as compared to NPC derived from healthy twins are highlighted red. Genes, underexpressed in NPC derived from twins with PD, as compared to NPC derived from healthy twins are highlighted green.

**Figure 5 cells-10-03478-f005:**
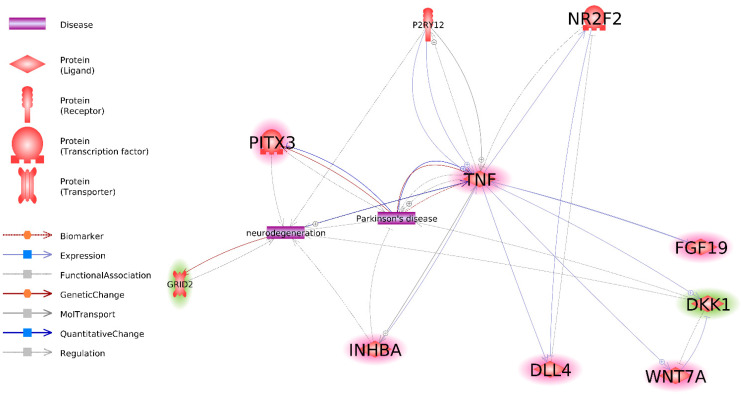
Interaction between most important DEG IPSC and DEG NP based on Pathway Studio data. Genes, overexpressed in NPC derived from twins with PD, as compared to NPC derived from healthy twins are highlighted red. Genes, underexpressed in NPC derived from twins with PD, as compared to NPC derived from healthy twins are highlighted green. Non highlighted genes are underexpressed in IPSC, derived from twins with PD as compared to IPSC, derived from healthy twins.

**Table 1 cells-10-03478-t001:** Top 3 DEGs up- and down-regulated by fold change in IPSC.

Ensembl_ID	Gene Symbol	FC (Gene Expression in IPSC, Derived from Twins with PD/Gene Expression in IPSC, Derived from Healthy Twins)	*p*-Value with FDR Adjustment
ENSG00000279483	AC090498.1	195	2.91 × 10^−4^
ENSG00000188985	DHFRP1	91	3.61 × 10^−3^
ENSG00000273213	H3-2	11	3.20 × 10^−2^
ENSG00000263798	AC018521.1	0.023	2.27 × 10^−3^
ENSG00000196350	ZNF729	0.016	1.13 × 10^−3^
ENSG00000263711	LINC02864	0.003	1.61 × 10^−4^

**Table 2 cells-10-03478-t002:** Top 5 DEGs up- and down-regulated by fold change in NPC.

Ensembl_ID	Gene Symbol	FC (Gene Expression in NPC, Derived from Twins with PD/Gene Expression in NPC, Derived from Healthy Twins)	*p*-Value with FDR Adjustment
ENSG00000279483	AC090498.1	264	4.96 × 10^−6^
ENSG00000214548	MEG3	162	2.99 × 10^−3^
ENSG00000181634	TNFSF15	105	2.99× 10^−3^
ENSG00000130300	PLVAP	31	1.38 × 10^−5^
ENSG00000181885	CLDN7	28	1.31 × 10^−3^
ENSG00000197705	KLHL14	0.033	2.02 × 10^−2^
ENSG00000022556	NLRP2	0.032	3.70 × 10^−3^
ENSG00000110077	MS4A6A	0.012	4.87 × 10^−4^
ENSG00000196109	ZNF676	0.012	1.64 × 10^−4^
ENSG00000263711	LINC02864	0.002	1.33 × 10^−8^

**Table 3 cells-10-03478-t003:** Significantly enriched GO BP terms for DEG NPC.

GO Term	GO Group	Number Of DEG, Associated with Term	Percentage of DEG among All Genes, Associated with the Term	Bonferroni Adjustment *p*-Value of Hypergeometric Test for Enrichment
nervous system development	I	60	18.93	6.39 × 10^−3^
neurogenesis		46	22.89	3.78 × 10^−4^
generation of neurons		41	22.40	2.88 × 10^−3^
neuron differentiation		37	22.70	6.29 × 10^−3^
anatomical structure morphogenesis	II	88	19.13	2.10 × 10^−5^
tube development		37	23.57	2.47 × 10^−3^
tube morphogenesis		33	24.26	4.34 × 10^−3^
animal organ morphogenesis		46	23.96	8.95 × 10^−5^
regulation of metabolic process	III	202	14.09	3.52 × 10^−3^
cellular biosynthetic process		167	15.24	2.42 × 10^−4^
organic substance biosynthetic process		169	15.21	2.22 × 10^−4^
regulation of cellular metabolic process		174	14.49	4.23 × 10^−3^

**Table 4 cells-10-03478-t004:** Relation between DEG NPC, involved in groups I and II of GO BP with *TNF* based on data, found in literature.

Gene	Protein	Protein Function	Direction of Interaction with TNF (Gene or Protein)	Expected Direction of Expression Change in NPC, Derived Twins with PD as Compared to NPC Derived Healthy Twins Based on Overexpression of TNF in Litrearure Data	Observed Direction of Expression Change in NPC, Derived from Twins with PD as Compared to Healthy Twins	Relation with PD
*DLL4*	DLL4	Transcription factor regulating the evolutionarily conserved Notch pathway, which plays a significant role in cellular differentiation, proliferation, and apoptosis [67].	TNF inhibits *DLL4* [68,69]	↓	↑	Not shown previously
*FGF19*	FGF19	FXR receptor ligand playing a role in processes such as protein synthesis and carbohydrate/lipid metabolism [70].	TNF inhibits *FGF19* [71]	↓	↑	Not shown previously
*INHBA*	Activin-A	Member of the family of TGF-beta factors playing a role in inflammation, fibrosis, and immunoregulation [72].	TNF induces *INHBA* [73,74,75,76,77,78,79]	↑	↑	In vivo [80]
*WNT7A*	Wnt7a	Activates the canonical and non-canonical Wnt signaling pathways. The Wnt pathway and this protein in particular are involved in multiple biological processes, including differentiation, proliferation, wound healing, and inflammation suppression [81,82].	TNF induces *WNT7A* [83,84]	↑	↑	Through canonical Wnt signaling [85]
*DKK1*	DKK1	Antagonist of the canonical Wnt pathway [86].	TNF inhibits [87,88] or induces [89,90,91,92,93] *DKK1* based on tissue	↑↓	↑	In vivo [94]

## Data Availability

Raw and processed data can be accessed in Gene Expression Omnibus, accession number GSE185009.

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
