# Peer review of "Transcriptome Analysis of Induced Pluripotent Stem Cells and Neuronal Progenitor Cells, Derived from Discordant Monozygotic Twins with Parkinson’s Disease"

_cells, 2021, doi:10.3390/cells10123478_

Round 1

Reviewer 1 Report

In this paper, the authors reported a RNAseq-based study on iPSCs and iPSC-derived NPCs from PD discordant monozygotic twins, which provides a bunch of implicative transcriptomic data that would be useful in deciphering the molecular basis of PD pathogenesis. This is a very innovative and interesting study, which may open up new ideas for new PD biomarker discovery with clinical transformation potential. From the perspective of academic criticism, several technical concerns need to be addressed to further improve the quality of this manuscript, as appended below.

The iPSC-derived NPCs were not purified after the differentiation. What is the ratio of NPCs in the population subjected to RNA-seq? A flow cytometry or IHC data will be necessary to address this.

To present the analysis, the comparison (like “NPCs from healthy donor” vs “NPCs from PD patient”) and the regulation type (“up-regulation” or “down-regulation”) need to be clearly shown in the figures or table.

For better visualization, the DEGs should be sorted by up-regulation and down-regulation. A list of top regulated genes in both cases should be extracted from the long list and included in the main figure.

The data would be more comprehensively present if the author include a gene set enrichment analysis (GSEA) for the existing dataset. (Subramanian, Aravind, et al. "Gene set enrichment analysis: a knowledge-based approach for interpreting genome-wide expression profiles." Proceedings of the National Academy of Sciences 102.43 (2005): 15545-15550.) The GSEA would be a complementary to the GO analysis.

Author Response

Dear Reviewer, 

Please see the response in the attachment below.

Reviewer 2 Report

In this article, the authors have investigated iPSCs and neuronal precursor cells (NPCs) derived from two pairs of monozygotic twins discordant for PD. RNAseq has been performed from iPSCs and NPCs in triplicates and was analyzed for DEGs and differentially regulated pathways comparing the unaffected and affected twins. Data analysis revealed 20 DEGs in iPSCs and 1906 DEGs in NPCs. Moreover, the authors provide results from pathway analyses and show changes for specific gene interactions / biological processes.

While the study has been executed in a straightforward manner, and data shown are indeed novel and could indicate gene expression differences related to PD, there is the necessity to include more data and further explanation. The study would benefit from the following additions/modifications:

  • Data are missing that show the pluripotent potential of the newly reprogrammed iPSC lines and the confirmation of a normal karyotype.
  • Data are missing to show that NPCs from the different lines/differentiations are comparable with each other regarding the expression of general neuronal markers, such as TUJ1, PAX6, MAP2… This information could be derived from the RNAseq data.
  • Have the twins been investigated for mutations in known PD genes?
  • The differential expression of the identified main candidates (Figure 5) needs to be validated by RT-PCR.
  • It seems inappropriate to claim that TNF is a master regulator of PD in the abstract. This part must be rephrased carefully.

Minor points

  • Differential gene expression can be caused by mutations, genetic variants, or epigenetic changes. The presence of genetic variants is extremely rare when looking at monozygotic twins. Therefore, differential gene expression is most likely caused by epigenetic changes. As the reprogramming process can have an impact on epigenetic changes, there is a risk that the observed differential gene expression in iPSC-derived cells does not reflect the original epigenetic situation in the parental cells. This should be mentioned in the discussion as a limitation of the study.
  • In line 93+94, it is written that patient information can be found in Table S1. Table S1 of this manuscript contains a list DEGs.

Author Response

(The authors gave the same response as above.)

Round 2

Reviewer 1 Report

The revision is acceptable and the response is fair. The paper can be accepted for publication.

Author Response

Dear Reviewer,

Thank you very much for your suggestions. We believe they enhanced quality and readability of our paper.

Reviewer 2 Report

While the authors have addressed most of the points there are still some modifications I would like to suggest:

  • The characterization data of the iPSC lines should be moved into the supplements.
  • The expression data shown for the general neuronal markers TUJ1 and Nestin are acceptable. MAP2 levels are extremely high for one of the lines. However, it is possible that the levels are overall very low as MAP2 is not an early marker and the difference seen could result from a very small population of cells. Can the authors perform a comparison between MAP2 and Nestin regarding intensity/concentration?
  • The expression levels of the candidate genes need to be visualized in some way to being able to see if the effect is driven mainly by one line or one differentiation. Can the authors generate bar plots for the candidate genes similar to those made for the general neuronal markers? These data should go into the supplements as well. It would be also interesting to see if the candidate genes are expressed at relatively low or high levels and could be shown in comparison to Nestin as well.

Author Response

Dear reviewer,

Thank you for your suggestions, 

Please see our response in attachment below.
